# A Novel Fusion Approach Consisting of GAN and State-of-Charge Estimator for Synthetic Battery Operation Data Generation

**Kei Long Wong** [1,*] **, Ka Seng Chou** [1,2] **, Rita Tse** [1] **, Su-Kit Tang** [1] **and Giovanni Pau** [2,3,4]

1   Faculty of Applied Sciences, Macao Polytechnic University, Macao SAR 999078, China
2   Department of Computer Science and Engineering, University of Bologna, 40126 Bologna, Italy
3   Autonomous Robotics Research Center, Technology Innovation Institute (TII),
    Abu Dhabi P.O. Box 9639, United Arab Emirates
4   Samueli Computer Science Department, University of California, Los Angeles, CA 90095, USA
*   Correspondence: keilong.wong@mpu.edu.mo

**Abstract:** The recent success of machine learning has accelerated the development of data-driven lithium-ion battery state estimation and prediction. The lack of accessible battery operation data is one of the primary concerns with the data-driven approach. However, research on battery operation data augmentation is rare. When coping with data sparsity, one popular approach is to augment the dataset by producing synthetic data. In this paper, we propose a novel fusion method for synthetic battery operation data generation. It combines a generative, adversarial, network-based generation module and a state-of-charge estimator. The generation module generates battery operation features, namely the voltage, current, and temperature. The features are then fed into the state-of-charge estimator, which calculates the relevant state of charge. The results of the evaluation reveal that our method can produce synthetic data with distributions similar to the actual dataset and performs well in downstream tasks.

**Keywords:** synthetic data; lithium-ion battery; battery data; generative adversarial network; state-of-charge estimation; machine learning

## 1. Introduction

Electric vehicle (EV) development is accelerating due to concerns about air pollution. Lithium-ion (Li-ion) battery technology has been widely adopted as the primary power source of EVs because of its long lifespan, high energy density, low self-discharge rate, and lightweight characteristics [1]. During the operation of EVs, battery management is vital to ensure the safety of the users. A battery management system (BMS) is applied in EVs to perform various tasks, including state estimation, thermal control, and battery cell balancing [2]. State estimation is one of the crucial jobs among them to guarantee dependable functioning and increase battery life.

The state of charge (SOC) and state of health (SOH) are two main concerns for the state estimation in BMSs. The SOC represents the currently available charge level of the battery. It is essential to develop reasonable control mechanisms in order to preserve energy and avoid overcharge and overdischarge situations [3]. The SOH indicates the current health of the battery. It is usually defined by the battery's internal resistance increment or remaining capacity loss [4]. SOC and SOH estimation can be divided into three types of approaches: data-driven, model-based, and a hybrid method that fuses them [5,6]. The model-based method is mainly based on the equivalent circuit model (ECM) method and electrochemical and mathematical models that mainly focus on identifying model parameters via experiments [7]. The data-driven method uses big data technology to estimate the battery state by utilizing the battery's historical operation data. Unlike the model-based method,

the data-driven method does not analyze the complicated electrochemical behavior inside the Li-ion battery. This eases the implementation difficulty of the data-driven method.

Recently, the success of neural network technology in various domains encouraged the adoption of neural networks for data-driven battery SOC and SOH estimation. Numerous results for SOC and SOH estimation were presented in various studies [8,9]. The performance of neural network technology (especially deep learning) relies heavily on the amount of training data [10]. A model trained with insufficient data cannot comprehensively capture the battery dynamics related to its state changes. Despite this, there is a lack of publicly available Li-ion battery datasets. In particular, it is rare to obtain field testing data that can reflect the conditions of the battery during its actual operation [11]. Typically, data augmentation is used to address the problem of insufficient data in machine learning.

The operation data of Li-ion batteries used for estimation of the SOC and SOH is typically in the form of a time series. The traditional data augmentation methods, such as cropping, rotating, and flipping applied in the computer vision domain, are not applicable for time series data. Time series data may become noisy when using the aforementioned traditional methods since they directly transform the original data. Synthetic data generation, which learns the original data distribution and creates new data that mimic the distribution, is a better option [12].

The generative adversarial network (GAN) has recently achieved great success in producing synthetic data for computer vision [13]. The field of synthetic data generation research using GANs is well studied, such as the cGAN [14] for assigning conditions to generated data, StackGAN [15] for text-to-image synthesis, and CycleGAN [16] in image-to-image translation. Although the majority of GAN research focuses on creating images, time series generation is also being investigated. As summarized by Brophy et al. [17], the application of a time series GAN includes data augmentation, imputation, denoising, and differential privacy. Inspired by the success of the time series GAN for synthetic data generation, utilizing a GAN for Li-ion battery operation data generation is proposed in this research.

In this work, we propose a novel approach that combines a generation module and an SOC estimator. A GAN-based generation module is employed to create synthetic battery operation data during the discharge phase. It is necessary to generate data from the discharge phase because the operation of the battery during discharge is more dynamic and variable than during charging. The synthetic battery operation data include the voltage, current, and temperature, all of which vary during operation. Using the data generated by the GAN-based generation module, a long short-term memory (LSTM)-based SOC estimator is applied to compute the corresponding SOC associated with the discharge operation. Therefore, we can generate synthetic battery operation data with four features: the voltage, current, temperature, and SOC. The main contributions are summarized as follows:

- The existing GAN and non-GAN methods for producing synthetic battery operation data are reviewed.
- We propose a fusion method for producing synthetic battery operation data during the discharge phase, which involves a GAN-based generation module combined with an LSTM-based SOC estimator.
- Qualitative and quantitative examinations are performed to compare the quality of the produced data to those of other time series GAN techniques.

The rest of this paper is organized as follows. A literature review of the related topics is presented in Section 2. Then, Section 3 explains the method used in this work. In Section 4, we evaluate the proposed method. Finally, Section 5 summarizes this paper and provides the concluding remarks.

## 2. Related Work

In this section, the fundamental of the GAN, its variation for time series data generation, and the current advances in GANs and related networks in the area of Li-ion batteries

are introduced. Furthermore, the present methodologies for generating synthetic battery operation data are described. Following that, we present the motivation for our proposed method based on the relevant works.

### 2.1. Generative Adversarial Network

The generative adversarial network (GAN) is an algorithm for training generative models proposed by Goodfellow et al. [18]. Fundamentally, a GAN consists of two neural networks: a generator and a discriminator. Training is conducted in the form of a zero-sum game between the two networks. The GAN architecture is shown in Figure 1. The generators attempt to create synthetic data that can be misinterpreted by discriminators as real data. In contrast, the discriminator is trained to distinguish between synthetic and real data. The primary goal of the GAN is to produce synthetic data that are indistinguishable from genuine data. The GAN has received tremendous attention since its debut in 2014. A tremendous amount of advanced architectures were proposed in the research field. Several of them are introduced in the following paragraph.

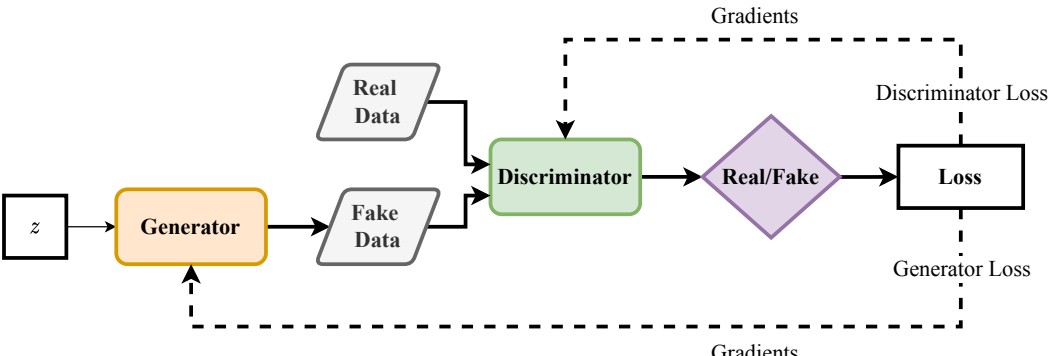

**Figure 1.** Generative adversarial network.

The conditional GAN (cGAN) [14] extends the GAN by including conditions. The condition is usually the desired class or label of the generated data. The training process of the cGAN is similar to that of the GAN, and it adds the condition in the input of the generator and discriminator. The generator creates synthetic data that are associated with the desired condition, and the discriminator identifies the data under the corresponding condition. As the associated label of the generated data is known, it can be useful in augmenting training data for classification tasks. The deep convolutional GAN (DCGAN) [19] combines deep convolutional neural networks and GANs that increase the training stability. The authors proposed several tricks to stabilize the training of GANs, including (i) replacing pooling with strided convolution so that the network learns its own spatial downsampling, (ii) using batch normalization in both the generator and discriminator to stabilize the training process, (iii) replacing the fully connected layers with a convolutional layer, (iv) the generator using ReLU for activation of all layers except the output layer, which should use Tanh, and (v) all layers in the discriminator using LeakyReLU. The InfoGAN [20] separates the input of the generator into $z$ (noise) and $c$ (code) and introduces a classifier $Q$ to determine the corresponding code of the output. The idea is to allow the generator to produce synthetic data associated with the code. The code may contain conditional information such as image rotation or the text character width. The Wasserstein GAN (WGAN) [21] improves the stability of training the GAN model, and it can avoid mode collapse and vanishing gradients in GANs. The ideas include (i) removing sigmoid activation from the last layer of the discriminator, (ii) the loss of the generator and discriminator does not taking a logarithm, and (iii) substituting the Wasserstein distance for binary classification in the discriminator. Moreover, Gulrajani et al. [22] proposed the WGAN with a gradient penalty (WGAN-GP) to further improve the WGAN by introducing a gradient penalty.

As previously indicated, most GAN applications are geared toward producing images. Brophy et al. [17] summarized that the time series-based GAN can be divided into discrete-time and continuous-time forms, each of which has different challenges. The battery operation data we study in this research are in continuous-time form; that is, a data value capture happens at every moment. There are primary concerns regarding the model's ability to capture the complexity between the temporal features and attributes for continuous-time data. The relationship between data features should be reflected in the generated data. For instance, the linkage between the current and voltage due to ohmic loss and the diffusion voltage in battery operation data [23] should be shown. In the research field, many works were presented to address the time series GAN problem. Some of them are reviewed in the following paragraph.

The time-GAN is a time series data generative model proposed by Yoon et al. [24]. The main idea is to combine the versatility of unsupervised training in GANs with the conditional probability principles of supervised autoregressive models so that the temporal dynamics is preserved. Li et al. [25] proposed a time series GAN model (TTS-GAN) based on the transformer encoder architecture [26]. The authors view the time series data as image data and process them with the transformer architecture. The time series sequence length is set to be the width of an image with a height of one, and the features of the time series data are treated as the channels of an image. Pei et al. [27] presented (RTSGAN) the use of GAN and an autoencoder combined to deal with real-world time series data that are usually characterized by a variable length, long sequence length, and missing values. Autoencoders are trained on real data with an encoder that encodes data to latent representations and a decoder that decodes the latent representations back into the original data. A GAN model involves training a generator to generate latent representations and feed both the real and synthetic latent representations to a discriminator to determine their authenticity. In order to generate synthetic data, the trained decoder decodes the generated latent representation.

Recently, a growing number of Li-ion battery-related studies have adopted GANs and other similar techniques. For instance, Zhang et al. [28] proposed the use of a GAN-conditional latent space (CLS) to generate pseudo-experimental Li-ion battery data. In their research, the battery charge and discharge characteristics were converted into image representation, and then the GAN-CLS was used to produce more relevant image data. The produced data were then applied to train a bidirectional-LSTM model for open circuit voltage characteristic prediction. Moreover, Gayon-Lombardo et al. [29] presented a DCGAN to generate 3D multi-phase electrode microstructural data. Furthermore, Hu et al. [30] proposed forecasting the calendar aging of Li-ion batteries through the use of a GAN designed based on the battery's electrochemical knowledge. In addition, Faraji Niri et al. [31] proposed using an auto-encoder decoder for microstructure reconstruction of Li-ion battery electrodes.

### 2.2. Synthetic Battery Operation Data

Successful data-driven estimation of a battery's state requires a variety of large battery datasets. A battery dataset that covers a greater number of situations is more likely to be convincing. It is still difficult to find sufficient public datasets that cover a variety of scenarios related to battery operation. In response to the problem of lacking data, synthetic data generation has become an option. In synthetic data generation, artificial data that have a similar distribution to the original data are generated to expand an existing dataset. A review of several studies relating to the generation of synthetic battery operation data is presented below.

Pyne et al. [32] proposed a Markov chain- and neural network-based approach for generating synthetic battery operation data with SOH variation. They proposed generating feature data, namely the voltage centroid value (average value of voltage clustered by k-means), rather than generating the entire charge or discharge cycle with data sampled at every time step. In their framework, a synthetic current profile is first generated based on

Markov chain propagation. Then, a neural network trained by real data is used to estimate the voltage centroid value with the generated current profile. Time series GAN-based synthetic battery operation data generation was proposed by Naaz et al. [33]. Their results were evaluated by the Oxford battery degradation dataset [34] and NASA prognostics dataset [35]. The synthetic data of both datasets were found to have similar distributions to the original data. As a result of concerns regarding the intensive computing requirement of the GAN, Naaz and Channegowda [36] proposed generating synthetic battery data by using a recurrent neural network (RNN). The basic idea is to train the model to forecast future voltage, current, and capacity values in order to generate new data.

### 2.3. Motivation for the Proposed Method

The existing literature shows the possibility of expanding battery datasets. However, there are still limitations in synthetic battery data generation research. As the feature-based generation method [32] only generates the selected features of the time series data, different types of applications, such as forecasting the battery temperature, cannot be performed on the feature-based data. Furthermore, the diversity of the synthetic data created by projecting future operations [36] is limited since the generated data are confined by the previous operation. What is more, the GAN-based approach [33] exhibits close distribution between synthetic and real data. However, the synthetic voltage and SOC value show high jitter compared with the real data. In addition, the battery data generated using the present time series GAN approaches lack the essential battery operating characteristic, specifically the SOC value, which is anticipated to decline progressively over the discharge cycle.

To address the issues highlighted above, we suggest a fusion strategy that combines a generation module with an SOC estimator. The GAN-based generation module generates current, voltage, and temperature values that mimic actual discharge operation. Based on these synthetic operation data, an SOC estimator, which was validated to be able to estimate the SOC accurately [37], can be used to produce the associated SOC value. Using an SOC estimator to generate the SOC value rather than creating all values using a GAN can result in synthetic data with SOC drops that are more similar to actual battery operation.

## 3. Methodology

This section introduces the dataset utilized in our research. The suggested generative framework and its components are then described in depth. Finally, the suggested models' training procedures are described.

### 3.1. Li-Ion Battery Dataset

In this research, a public dataset of the LG 18650HG2 Li-ion battery (LG dataset) is used. This dataset was published by Kollmeyer et al. [38] and made available to the public through Mendeley Data. Charge and discharge tests at various temperatures were conducted to collect the data. The discharge cycles followed the driving cycles, namely UDDS, LA92, US06, etc. and a combination of these cycles. Throughout this study, we focus on the discharge cycle at positive temperatures since these represent the majority of use cases. Specifically, in our experiments, we used discharge cycles at 0 °C, 10 °C, and 25 °C. The data were collected in the form of time series with voltage, current, temperature, and SOC values sampled at a 10 Hz rate. An example discharge cycle from the dataset is shown in Figure 2.

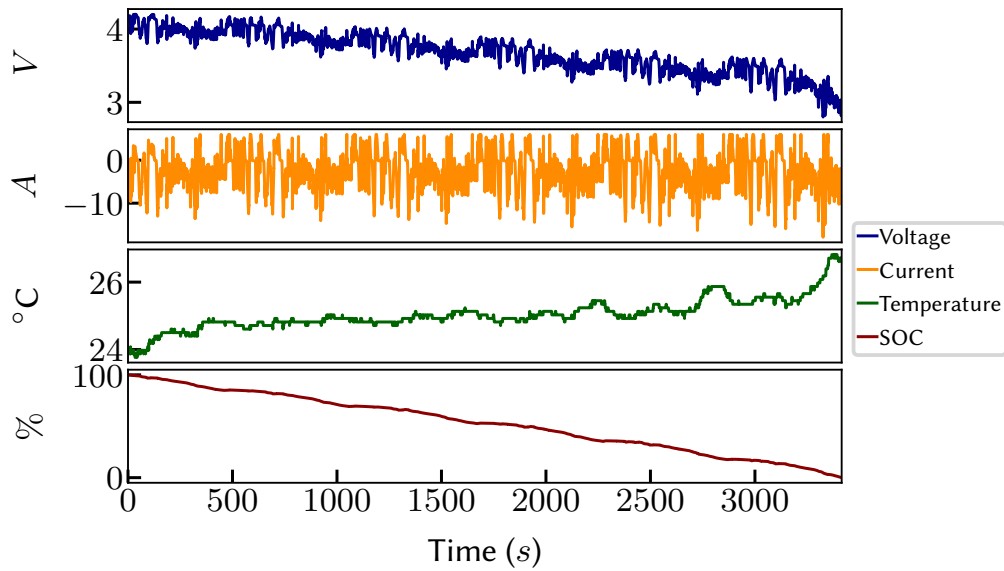

**Figure 2.** LG 18650HG2 Li-ion battery dataset example.

### 3.2. Generative Framework

The generative framework consists of two main components: a GAN-based generation module and a deep LSTM-based SOC estimator. The two components are trained separately. In both components, training data are drawn from the LG dataset. The GAN-based generation module employs a generator and discriminator that are trained together to create high-fidelity synthetic data for the voltage, current, and temperature during a discharge cycle. The SOC estimator is trained to estimate the SOC based on the input voltage, current, and temperature value. By combining the synthetic battery operation data from the generator with the estimated SOC derived from the synthetic data, we can generate a synthetic Li-ion battery operation dataset.

Figure 3 illustrates the overview of the proposed framework. In the figure, $V$ represents the voltage, $I$ indicates the current, $T$ is the temperature, and $z$ is the latent space sampled from the standard normal distribution. The orange dashed line represents the SOC estimator training phase, the blue dashed line indicates the GAN training phase, and the red line represents the synthetic data generation phase. The directions of the arrows point out the input and output of the corresponding components in different phases. It is worth mentioning that the latent space $z$ is applied in both the GAN training phase and the synthetic data generation phase. The generator is subjected to generating synthetic data from random Gaussian noise both during training and during production. The details of each component and the training process are described in the following sections.

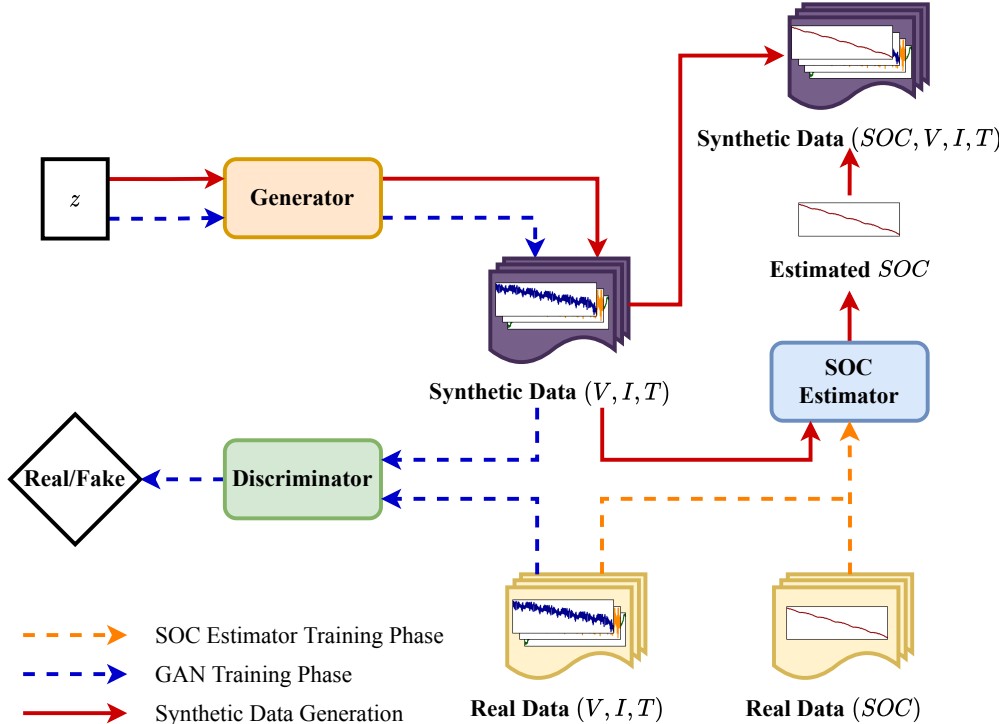

**Figure 3.** Generative framework overview. The orange dashed line indicates the training phase of the SOC estimator. The blue dashed line represents the GAN training phase. The red line indicates the synthetic data generation process.

### 3.3. GAN-Based Generation Module

The generation module consists of a GAN model combined with an LSTM-based generator and discriminator. The generator accepts latent space generated from a standard normal distribution in the form of an input matrix $z \in \mathbb{R}^{n \times t \times k}$, where $n$ denotes the batch size, $t$ denotes the time step length, and $k$ denotes the input dimension. The input latent space is passed through a stacked LSTM model to produce the voltage, current, and temperature in the form of a matrix $\tilde{x} \in \mathbb{R}^{n \times t \times 3}$. The input dimension of the generator is 128. On the other hand, the discriminator accepts a sequence of the voltage, current, and temperature to examine their authenticity with a stacked LSTM model. The architectures of the generator and the discriminator are shown in Figure 4.

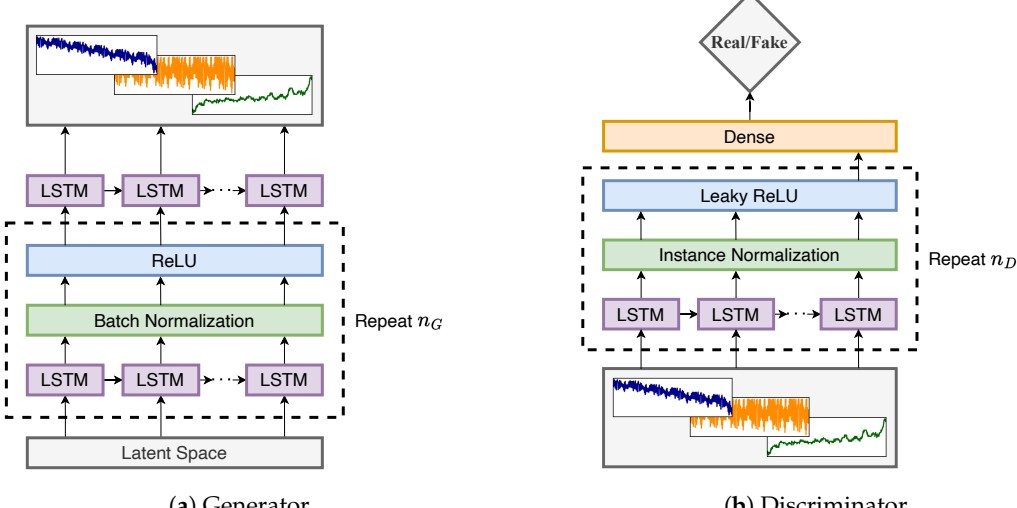

(**a**) Generator    (**b**) Discriminator

**Figure 4.** GAN-based generation module architecture.

As shown in the figure, the stacked LSTM part in the generator is repeated $n_G$ times, and an LSTM layer is used as the output layer directly. The LSTM layer in the stacked part is followed by batch normalization and a ReLU activation function. In our experiment, we set $n_G = 4$, and the number of features in the hidden state of LSTM was 1024, 512, 256, and 128. In the discriminator, the stacked LSTM part is repeated $n_D$ times, and the output of its last step is passed to a fully connected layer to produce the critic. After each LSTM layer, an instance normalization layer and a leaky ReLU activation function are applied. As in the generator, we set $n_D = 4$, and the number of features in the hidden state of LSTM was 128, 256, 512, and 1024.

As a point of clarification, this work focuses on generating 300-step-long sequential data (equal to 30 s). In order to be able to produce a 300 step SOC sequence, the time step of the generator output and the input of the discriminator was set to be 600 (60 s), since the SOC estimator we used had a many-to-one set-up.

### 3.4. Generation Module Training

The training of the generation module followed the improved version of the WGAN (WGAN-GP) [22] framework. A min-max adversarial loss function was used to train both the generator and discriminator networks simultaneously. The objective function of the generation module is defined as

$$\min_{G} \max_{D} \mathbb{E}_{x \sim \mathbb{P}_x}[D(x)] - \mathbb{E}_{z \sim \mathbb{P}_z}[D(G(z))] \tag{1}$$

where $D$ denotes the 1-Lipschitz discriminator, $G$ denotes the generator, $\mathbb{P}_x$ is the real data distribution, and $\mathbb{P}_z$ is the latent space distribution (standard normal distribution in our case). Through iterative training of the 1-Lipschitz discriminator and generator, the generator is trained to minimize the 1-Wasserstein distance between the real and synthetic data distribution. To improve the stability of training, the gradient penalty is used to replace the weight clipping strategy in the WGAN. A gradient penalty is added to the loss and defined as

$$\lambda \mathbb{E}_{\hat{x} \sim \mathbb{P}_{\hat{x}}}\left[\left(\|\nabla_{\hat{x}} D(\hat{x})\|_2 - 1\right)^2\right] \tag{2}$$

where $\lambda$ is the gradient penalty coefficient and $\hat{x}$ is given by

$$\hat{x} \leftarrow \epsilon x + (1 - \epsilon)G(z) \tag{3}$$

with the random number $\epsilon$ sampled from the continuous uniform distribution.

In total, we trained the generation module for 20,000 epochs with a batch size of 64. During each epoch, the discriminator was trained five times, and the generator was trained once. The gradient penalty coefficient $\lambda$ was set to be 10. The Adam optimizer [39] was used in both the generator and discriminator networks with a learning rate of 0.0001, $\beta1$ as 0.5, and $\beta2$ as 0.999. The networks were implemented in PyTorch [40], and all the training was performed on the DGX station with Tesla V100 GPUs. The source codes of the implemented networks have been made available in an online repository reachable at https://github.com/KeiLongW/synthetic-battery-data (accessed on 5 January 2023).

### 3.5. Deep LSTM-Based SOC Estimator

The SOC estimator is a deep LSTM-based model proposed in [37]. The use of LSTM over an RNN makes the model more resistant to problems related to gradient exploding and vanishing. An LSTM cell contains multiple components, including a forget gate, input

gate, output gate, cell state, and hidden state. The calculation steps of an LSTM cell are shown in Equation (4):

$$
\begin{aligned}
f_t &= \sigma(W_x^f x_t + W_h^f h_{t-1} + b^f) \\
i_t &= \sigma(W_x^i x_t + W_h^i h_{t-1} + b^i) \\
\tilde{c}_t &= tanh(W_x^c x_t + W_h^c h_{t-1} + b^c) \\
c_t &= f_t \odot c_{t-1} + i_t \odot \tilde{c}_t \\
o_t &= \sigma(W_x^o x_t + W_h^o h_{t-1} + b^o) \\
h_t &= o_t \odot tanh(c_t)
\end{aligned}
\tag{4}
$$

where $f$ is the forget gate, $i$ is the input gate, $o$ is the output gate, $c$ is the cell state, $h$ is the hidden state, $\sigma$ is the sigmoid function, $\odot$ is the Hadamard product, $W$ is the weight matrix, $x$ is the input vector, and $b$ is the bias. The first step is to determine what information should be forgotten from the previous cell state. Following this, it is determined whether the information will be stored in the cell state. The next step is to combine the current cell state with the previous cell state. As a final step, the output gate with the sigmoid function determines which part of the cell state should be propagated to the hidden state.

In this work, we adopt the set-up of 300 time steps for the input sequence length; that is, the input fed to the network is a time series data with a length of 300. The features of the time series data are the voltage, current, and temperature. The network estimates the *SOC* value at the last step. The input $x$ at a time step $t$ can be described as follows:

$$
x_t = [V_t, I_t, T_t]
\tag{5}
$$

where $V$ is the voltage, $I$ is the current, and $T$ is the temperature. The input sequence is defined as follows:

$$
X = [x_1, x_2, ..., x_n]
\tag{6}
$$

where $n = 300$ in our case. Additionally, the output can be described as follows:

$$
Y = SOC_n
\tag{7}
$$

where $Y$ is the *SOC* value at the last step $n$. The aforementioned set-up is a many-to-one method. The model architecture consists of two LSTM layers followed by three fully connected layers. The architecture of the SOC estimator is depicted in Figure 5.

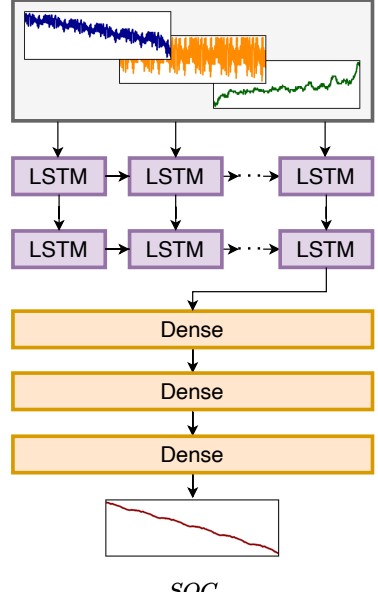

**Figure 5.** SOC estimator architecture.

The number of cells of the LSTM layers was 256, and the number of neurons for the dense layers was 256, 128, and 1. In this research, instead of retraining the SOC estimator, the trained weights of the SOC estimator provided in a public repository at https://github.com/KeiLongW/battery-state-estimation/releases/tag/v1.0 (accessed on 5 January 2023), were used in the synthetic data generation stage.

## 4. Result and Discussion

In this paper, we evaluate the proposed method (LSTM-GAN + SOC estimator) using qualitative and quantitative approaches. The proposed method is compared to two GAN-based time series data generators, namely the RTSGAN [27] and our generation module without the SOC estimator (LSTM-GAN). It is important to note that all three methods were trained with 20,000 epochs so that the comparison could be carried out fairly.

### 4.1. Qualitative Evaluation

In qualitative evaluation, we visually compare the created synthetic data to the real data. Then, we apply principal component analysis (PCA) [41] and t-distributed stochastic neighbor embedding (t-SNE) [42] on both the synthetic and actual data to examine their data distributions. Furthermore, by visualizing the created data throughout epochs, we show how the training iterations increase the quality of the generated synthetic data.

#### 4.1.1. Data Visualization

Once the training of the LSTM-GAN was completed, synthetic battery operation data containing the voltage, current, and temperature could be generated by supplying Gaussian noise to the generator. Then, we applied them to the SOC estimator to form a sequence of the voltage, current, temperature, and SOC data. For another two methods, the generator created all four features at once. Figure 6 demonstrates the examples of real data and the generated data from the three methods.

In the figures, the blue line indicates the voltage value, the orange line indicates the current value, the green line indicates the temperature value, and the red line represents the SOC value. All the features were rescaled by minimum-maximum normalization so that all of them ranged from 0 to 1.

The real data (Figure 6a) clearly demonstrates that the voltage and current curves were subject to fluctuations. This was primarily due to the fact that the discharge curve in the dataset was generated using a simulated driving profile that was loaded with varying discharge currents. Additionally, there were small jitters in the temperature curve. In contrast, the SOC curve was relatively smooth, since it was expected that the SOC change would be small within a short period of time (30 s). The synthetic data generated by the RTSGAN (Figure 6b), however, did not exhibit fluctuations in the voltage or current. A flat and smooth pattern was present in all of the four features in the synthetic data generated by the RTSGAN. The synthetic data created by LSTM-GAN-based methods (Figure 6c,d), on the other hand, displayed oscillating voltage and current curves. The temperature curve also exhibited rather minor jitters; that is, the voltage, current, and temperature curves generated by the LSTM-GAN-based methods were closer to the actual discharge cycle. However, the SOC curve created by LSTM-GAN without the SOC estimator fluctuated as well. As previously stated, the SOC value was not expected to fluctuate much in a short period of time. We can notice a relatively smoother SOC curve in the data when we employed the SOC estimator (Figure 6d). The proposed method exhibited oscillations in the voltage, current, and temperature curves while maintaining the smoothness of the SOC curve.

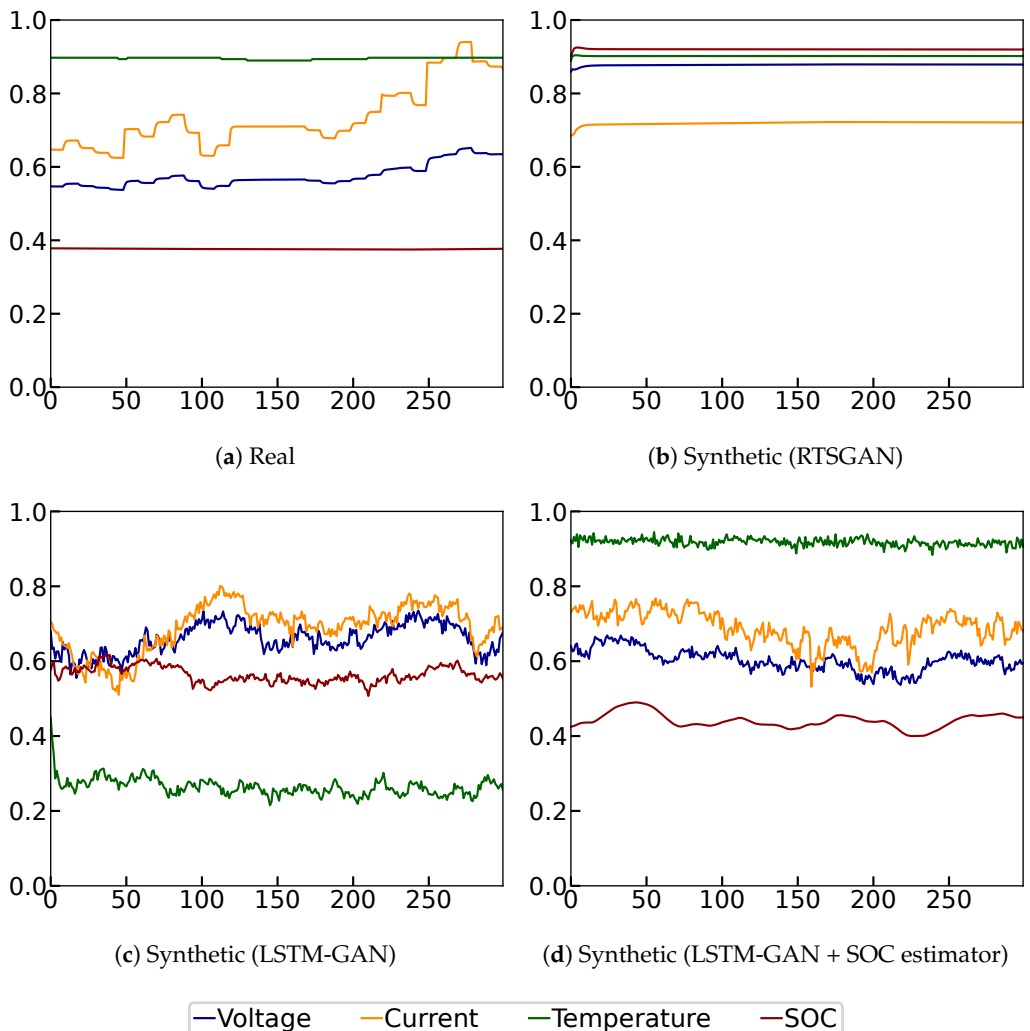

**Figure 6.** Data example of the real and synthetic battery operation data. The four features (voltage, current, temperature, and SOC) are rescaled by minimum-maximum normalization so that they have the same common scale.

### 4.1.2. Data Distribution

We analyze the data distribution of the real data and synthetic data by using PCA and t-SNE. A sequence of battery operation data with four features was reduced to two-dimensional data using PCA and t-SNE, allowing us to visualize their distribution in two dimensions. The outputs of PCA and t-SNE were generated by the machine learning library sklearn [43]. The PCA results are shown in Figure 7, and the t-SNE results are depicted in Figure 8.

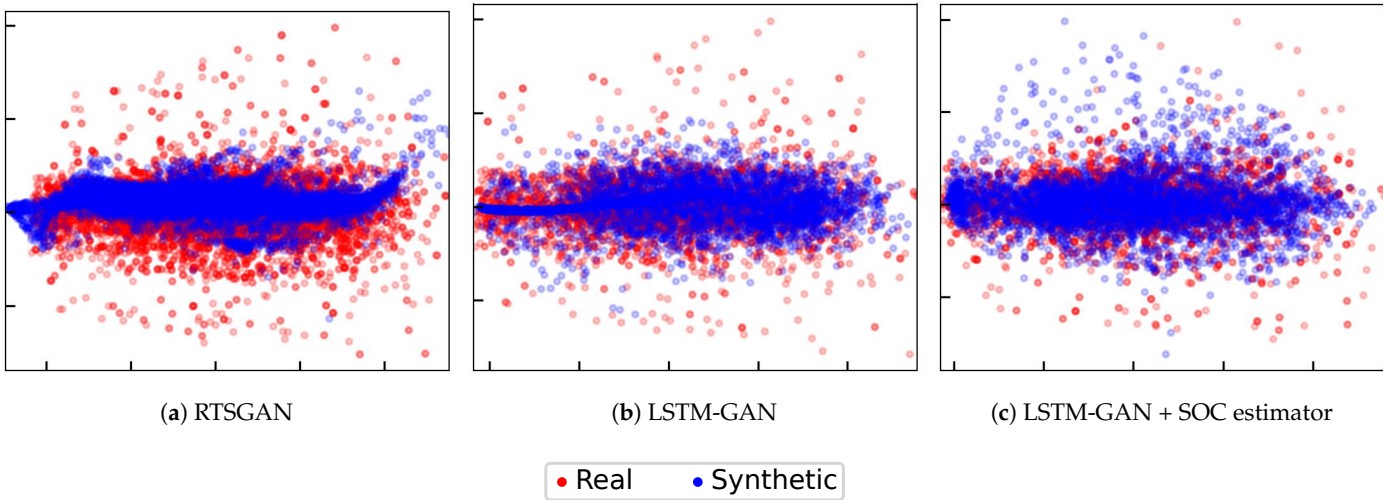

(**a**) RTSGAN　　　　　(**b**) LSTM-GAN　　　　　(**c**) LSTM-GAN + SOC estimator

● Real　　　● Synthetic

**Figure 7.** PCA results of the real data and synthetic data generated by the three generators.

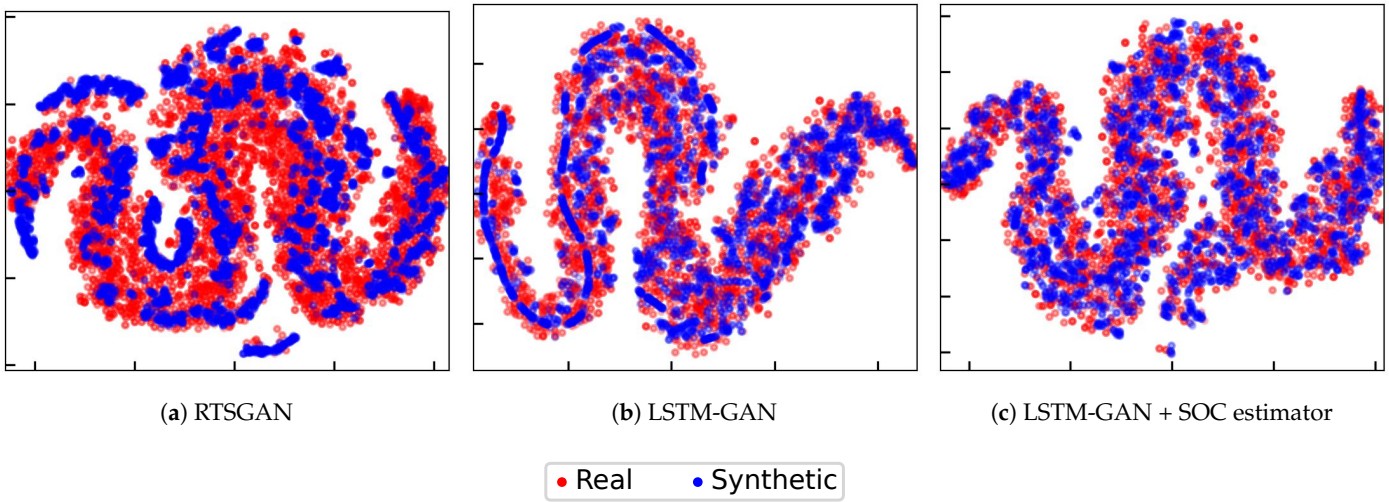

(**a**) RTSGAN　　　　　(**b**) LSTM-GAN　　　　　(**c**) LSTM-GAN + SOC estimator

● Real　　　● Synthetic

**Figure 8.** t-SNE results of the real data and synthetic data generated by the three generators.

Every point in the figure represents a sequence of mean values for the four features (voltage, current, temperature, and SOC). The red points indicate the real data, and the blue points are the generated data. If the data points overlap, then it is considered that the distribution is close. Using their overlap areas, we were able to evaluate how well we could approximate the data distribution.

The synthetic data created by the three techniques matched the real data distribution, as demonstrated by the PCA and t-SNE findings. Furthermore, we can see that the LSTM-GAN-based approaches created more overlapping data (Figures 7b,c and 8b,c). This indicates that the LSTM-GAN-based approaches created more diverse synthetic data. One probable explanation is that they had more fluctuation in the produced data, as stated in Section 4.1.1.

### 4.1.3. Training Curve

With each iteration of the training, the quality of the data generated by the generation module was expected to improve. We present here the generated synthetic data, PCA, and t-SNE results along with the training iteration for evaluation purposes. Figure 9 shows the generator and discriminator loss with the corresponding generated samples.

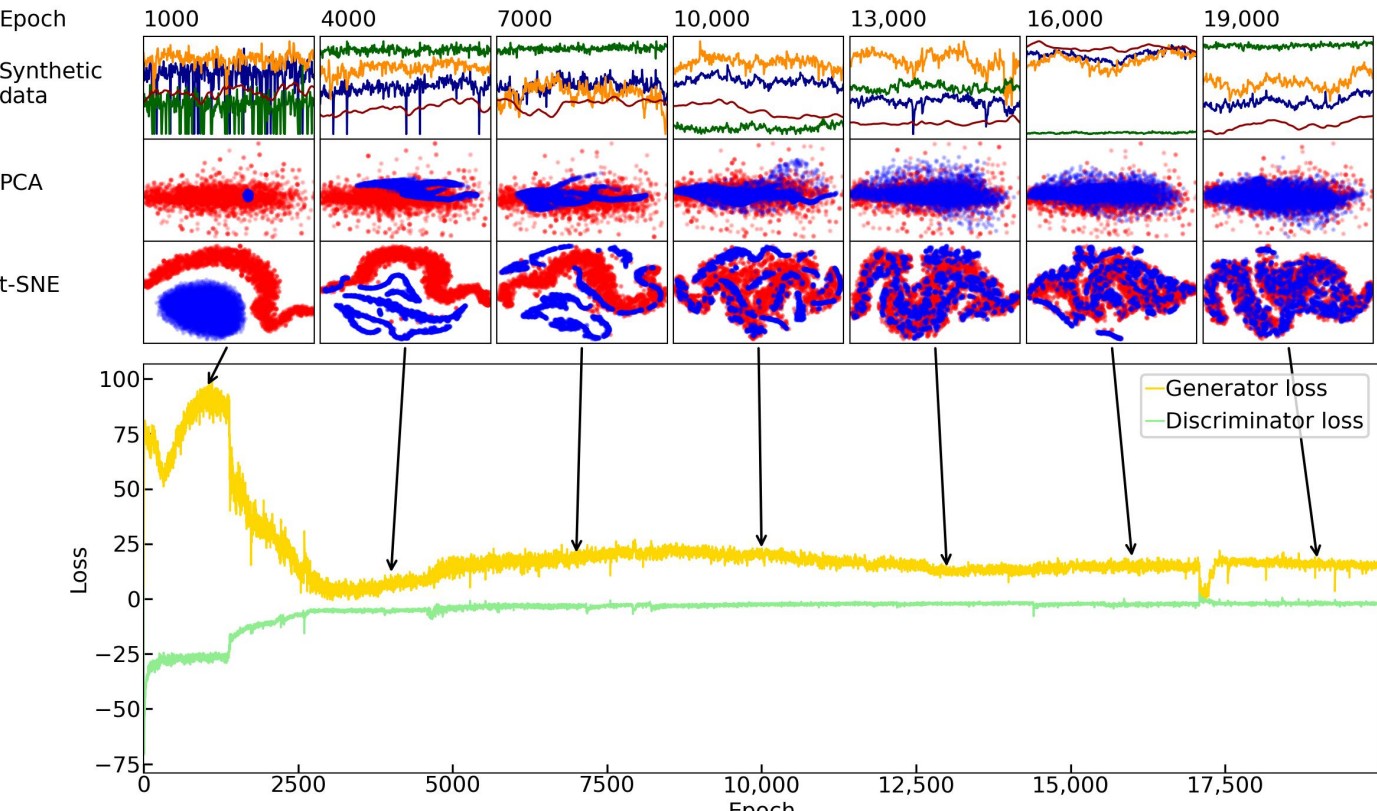

**Figure 9.** Training curve and the corresponding PCA, t-SNE, and synthetic data generated at different epochs of training.

The samples generated at epochs 1000, 4000, 7000, 10,000, 13,000, 16,000, and 19,000 are displayed in the figure. It is worth mentioning that we skipped the example from the last epoch (20,000) because we already demonstrated it in the above sections by using the result generated from the 20,000th epoch.

As observed from the training curve, the generator and discriminator are competing against each other. The better the performance for one, the greater the loss for the other. For instance, the discriminator's loss increases as the generator's loss decreases. At the beginning of the epoch (1000), the generator performed the worst. There was disorder in the generated synthetic data, and there was a lack of coherence in the distribution of the data. The generator loss became steady toward 7000 epochs, as seen in the loss curve and the generated synthetic data, PCA, and t-SNE.

It is clear that the quality of the generated data began to stabilize around epoch 10,000. In particular, the distribution between the synthetic and real data became closer in both the PCA and t-SNE, and the best result was observed from the samples of the 19,000th epoch. Although there were some tremors around epoch 17,500, the loss returned to a steady level after a short period of training. As a result, it is advised that the LSTM-GAN be trained with more than 10,000 epochs for a satisfactory outcome.

### 4.2. Quantitative Evaluation

Our quantitative evaluation employs the train on synthetic, test on real (TSTR) approach to validate the performance of the synthetic data in a voltage prediction downstream task. Moreover, an authenticity classifier is trained to determine whether the presented data are real or synthetic, and the performance of the classifier is used for evaluating the subjected synthetic data generator's performance.

4.2.1. Train on Synthetic, Test on Real

We assessed the quality of the generated synthetic data using the TSTR approach. A downstream model was trained using purely synthetic data and tested using purely real data. The task of the downstream model is simple many-to-many prediction of the voltage through a sequence of currents, temperatures, and SOC. The model is based on the neural network architecture. It consists of two fully connected layers with 64 hidden units and ReLU activation in the first layer. Here, we name this the dense model. We trained the dense model using the Huber loss [44] as a loss function and optimized it using the Adam optimizer [39] with a learning rate of 0.001. The performance of the downstream task was evaluated through the mean absolute error (MAE) and root mean square error (RMSE), which are given by

$$\text{MAE} = \frac{\sum\limits_{i=1}^{n} |y_i - \hat{y}_i|}{n} \tag{8}$$

$$\text{RMSE} = \sqrt{\frac{\sum\limits_{i=1}^{n} (y_i - \hat{y}_i)^2}{n}} \tag{9}$$

where $y$ is the true value and $\hat{y}$ is the predicted value. A more concrete evaluation was conducted by training and evaluating the model 30 times and averaging the results. The average performance of the dense model based on the synthetic data generated by different approaches is shown in Table 1.

**Table 1.** TSTR performance.

|                          | RMSE  | MAE   |
| ------------------------ | ----- | ----- |
| RTSGAN                   | 3.15% | 2.31% |
| LSTM-GAN                 | 3.36% | 2.55% |
| LSTM-GAN + SOC estimator | 2.83% | 2.08% |

As shown in the table, the model trained by the data produced by our proposed method outperformed the other methods. The model trained using synthetic battery data generated from LSTM-GAN + SOC estimator resulted in the lowest RMSE and MAE when testing with the real data.

4.2.2. Authenticity Classifier

To further evaluate the authenticity of the generated synthetic data, we trained an authenticity classifier. The classifier performed a many-to-one task that was used to determine if the supplied data were real or synthetic. It was fed with a sequence of voltage, current, temperature, and SOC data and output their corresponding label (real or synthetic) probabilities. By evaluating the classifier's performance for classification, we could verify the authenticity of the synthetic battery operation data generated. Poor performance by the authenticity classifier results in a higher degree of authenticity for synthetic data, as the classifier cannot distinguish between the real and synthetic data. The classifier is a fully connected model with two dense layers. The term dense classifier is used here. In the first layer, 32 hidden units were present, and the probability output from the last layer was derived from the sigmoid output activation. The classifier was trained with binary cross-entropy as the loss function and optimized with the Adam optimizer [39] with a learning rate of 0.001. The training and testing data of the classifier were a combination of the generated synthetic data and the real data. A random split of 70% training data and 30% testing data was performed. Then, the classifier was trained by 70% of the data and tested by 30% of the data. As with the testing in TSTR, we trained and evaluated the classifier 30 times and took the average of the result. Table 2 displays the average accuracy of the classifier against the synthetic battery operation data generated by the three generators.

**Table 2.** Authenticity classifier results.

|  | Accuracy |
|---|---|
| RTSGAN | 0.70 |
| LSTM-GAN | 0.66 |
| LSTM-GAN + SOC estimator | 0.65 |

The table displays the classifier's accuracy in detecting the authenticity of the provided sequence of data. In general, the lower the accuracy, the greater the quality of the synthetic data created as a result of the classifier's failure to distinguish between real and synthetic data. Based on the above results, it can be observed that using the data produced by the LSTM-GAN + SOC estimator performed the worst in classification. Consequently, our proposed method generated synthetic battery operation data that were the most accurate at replicating the original data.

*4.3. Discussion and Future Work*

In this research, it was demonstrated that the proposed LSTM-GAN + SOC estimator can produce battery operation data in which data distribution is closely related to the actual data and maintains both the variability and smoothness attributes. Nevertheless, the training of a deep GAN model and the generation of synthetic data are time-consuming. We proposed using deep LSTM for both the generation module and SOC estimator in this work. In our test environment, each epoch of GAN training took around 65 s, and full training took more than 350 h. For evaluation purposes, the generation of 4000 samples of voltage, current, and temperature data, as well as the estimate of the SOC, took around 3000 s in total. Despite the length of time consumed, it is uncommon to employ a synthetic data generation application in real time. The most typical use of our proposed approach is the augmentation of discharge profiles to an existing battery operation dataset.

We investigated expanding a battery's operating dataset containing the voltage, current, temperature, and SOC in the discharge scenario in this study. It would be interesting to investigate battery operation data with additional characteristics in the future, such as the SOH of the battery, or even expand the dataset taken from the battery pack during actual EV operation.

The produced synthetic battery operating data in this study were unconditioned; that is, the data were created at random from Gaussian noise. In the future, it would be beneficial to examine the cGAN [14] framework, which could conditionally regulate the created data. The condition might be, for example, the health of the battery cell, the driving situation when discharging the battery pack, or the ambient temperature of the discharging environment.

**5. Conclusions**

Data-driven Li-ion battery state estimation and prediction are significantly influenced by the amount of historical battery data available. The collection of field testing battery operation data is time-consuming and challenging. Data shortages can be addressed by creating synthetic data that mimics the original dataset. This paper presents a fusion method that combines a GAN-based generation module and an SOC estimator for synthetic battery operation data generation. The generation module and the SOC estimator are based on the deep LSTM architecture. The generation module creates voltage, current, and temperature data, which it then applies to the SOC estimator to construct a synthetic battery operation dataset with four attributes.

The proposed method was tested by a public Li-ion battery operation dataset sampled by a variable discharge profile. The evaluation results show that our proposed method can generate synthetic battery operation data with a data distribution close to the original dataset. Furthermore, the downstream task trained using the synthetic battery operation

data generated by our proposed method exhibited the best performance when tested on actual data.

In conclusion, this work demonstrates the feasibility of combining a GAN and SOC estimator for synthetic battery operation data generation. It is anticipated that this approach will benefit future data-driven battery state estimation and prediction research.

**Author Contributions:** Conceptualization, K.L.W.; methodology, K.L.W.; software, K.L.W.; validation, K.L.W.; formal analysis, K.L.W.; investigation, K.L.W.; resources, R.T., S.-K.T. and G.P.; data curation, K.L.W.; writing—original draft preparation, K.L.W.; writing—review and editing, K.L.W., K.S.C., R.T., S.-K.T. and G.P.; visualization, K.L.W.; supervision, R.T., S.-K.T. and G.P.; project administration, R.T., S.-K.T. and G.P.; funding acquisition, R.T., S.-K.T. and G.P. All authors have read and agreed to the published version of the manuscript.

**Funding:** This research received no external funding.

**Data Availability Statement:** Data are available from their original source cited in the article. The LG 18650HG2 Li-ion battery dataset is available at https://doi.org/10.17632/cp3473x7xv.3 (accessed on 5 January 2023).

**Acknowledgments:** This work was supported in part by the Macao Polytechnic University-Edge Sensing and Computing: Enabling Human-centric (Sustainable) Smart Cities (RP/ESCA-01/2020) and by the H2020 project titled "European Bus Rapid Transit of 2030: Electrified, Automated, Connected", EBRT-Grant Agreement No. 101095882.

**Conflicts of Interest:** The authors declare no conflict of interest.

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
