# Peer review of "A Novel Fusion Approach Consisting of GAN and State-of-Charge Estimator for Synthetic Battery Operation Data Generation"

_electronics, doi:10.3390/electronics12030657_

Round 1

Reviewer 1 Report

The litherature reviwe is not comprehensive addressing the recent developments regards GANs and simillar netwroks such as Auto-Encoder-Decoders in Lithium ion battery context. it is highly suggested to re-write the section addressing most highlighted reaserch of the time such as:

https://doi.org/10.1016/j.est.2020.101489

https://doi.org/10.1038/s41524-020-0340-7

https://doi.org/10.3390/en15124489

on the diagram of the proposed network on Figure 3, clarify the borders between data, GAN and The SoC estimator. do the arrows there point out the input output direction? I can quite understand why noise z is connected to generator via a sysnthetic data generation red arrow? it is really hard to interpret this diagram in my veiw.

also the notation z is both dedicated to noise and then later to the latent features which causes confusion.

It is highly recommended to be transparent regarding the hyper parameter tuning and also the configuration of such netwrok for reproducability, access to the source code via Github or a repository is required.

provde statements and if the access to the data set of this study is accessible by readers.

Author Response

Thanks for the comments. Please see below our response.

(1) The litherature reviwe is not comprehensive addressing the recent developments regards GANs and simillar netwroks such as Auto-Encoder-Decoders in Lithium ion battery context. it is highly suggested to re-write the section addressing most highlighted reaserch of the time such as: ...

Response: We extended the related work section to include some recent research of GAN and silmiar techniques in Li-ion battery context. (line 150~161)

(2) on the diagram of the proposed network on Figure 3, clarify the borders between data, GAN and The SoC estimator. do the arrows there point out the input output direction? I can quite understand why noise z is connected to generator via a sysnthetic data generation red arrow? it is really hard to interpret this diagram in my veiw.

Response: Yes, the arrows represent the input/output direction. The noise z is used in both training phase and generation phase because the generator generate data from random Gaussian noise both during training and production. We extended section 3.2 to clarify figure 3. (line 228~237)

(3) also the notation z is both dedicated to noise and then later to the latent features which causes confusion.

Response: They represent the same thing. We have updated the context related to the term 'noise' to avoid ambiguity.

(4) It is highly recommended to be transparent regarding the hyper parameter tuning and also the configuration of such netwrok for reproducability, access to the source code via Github or a repository is required.

Response: We added the reference to Github repository for open source purpose. (line 269)

(5) provde statements and if the access to the data set of this study is accessible by readers.

Response: We updated data availability statement.

Reviewer 2 Report

The authors propose a method to generate synthetic battery operation data in order to overcome the scarcity of public databases. Their proposal is based on data and a hybrid model, which combines a GAN to generate voltage, intensity and temperature data and a LTSM model to generate SOC data for the GAN generated data. This approach is meant to improve temporal consistency of generated SOC data and reduce jitter.

Visual evaluation using PCA and t-sne seem fair, but it could be possible to do research on synthetic data generation quality measures.

The comparative chart of the generated data between real and synthetic with different methods does not seem to reproduce the increase tendency of one of the signals. Perhaps this should be justified.

20.000 epochs with 64 batch size seem to be too long training for a network, specially after checking the evolution of the losses in Figure 9. It should be justified which criteria was selected to decide when to finish training. It seems that it was just manual visual inspection of t-sne and PCA plots. Is this right?

Author Response

Thanks for the comments. Please see below our response.

(1) Visual evaluation using PCA and t-sne seem fair, but it could be possible to do research on synthetic data generation quality measures.

Response: Apart from visual evaluation, we also include two quantitative methods (TSTR and Authenticity classifier) to evaluate the data.

(2) The comparative chart of the generated data between real and synthetic with different methods does not seem to reproduce the increase tendency of one of the signals. Perhaps this should be justified.

Response: When a discharge cycle is conducted, the load current is a variable signal, and we do not assume that there is a tendency for the load current to increase with time.

(3) 20.000 epochs with 64 batch size seem to be too long training for a network, specially after checking the evolution of the losses in Figure 9.

Response: In our paper, we recommend to train the GAN model more than 10000 epoch as it begins to stablize around epoch 10000. Although 20000 epochs required a long training time, it is uncommon to employ or re-train a synthetic data generation model in real time. The most typical usecase is to augment an existing dataset.

(4) It should be justified which criteria was selected to decide when to finish training. It seems that it was just manual visual inspection of t-sne and PCA plots. Is this right?

Response: Apart from visual inspection, we also save the sanpshot of the trained model at every 1000 epoch so that we can perform quantitative evaluation on every 1000 epoch of them.

Round 2

Reviewer 1 Report

I am happy with the responses